# Developmental Dysplasia of the Hip: Prevalence and Correlation with Other Diagnoses in Physiotherapy Practice—A 5-Year Retrospective Review

**DOI:** 10.3390/children9020247

**Published:** 2022-02-12

**Authors:** Veronika Vasilcova, Moqfa AlHarthi, Nadrah AlAmri, Peter Sagat, Peter Bartik, Ayman H. Jawadi, Martin Zvonar

**Affiliations:** 1Department of Kinantropology, Faculty of Sport Science, Masaryk University in Brno, Kamenice 753/5, 62500 Brno, Czech Republic; 2Pediatric Rehabilitation Department, King Abdullah Specialized Children Hospital, P.O. Box 22490, Riyadh 11426, Saudi Arabia; moqfa.alharthi@gmail.com (M.A.); Nadrah.alamri@gmail.com (N.A.); 3Health and Physical Education Department, Prince Sultan University, P.O. Box 66833, Riyadh 11586, Saudi Arabia; sagat@psu.edu.sa (P.S.); pbartik@psu.edu.sa (P.B.); 4College of Medicine, King Saud bin Abdul-Aziz University for Health Sciences, P.O. Box 22490, Riyadh 11426, Saudi Arabia; dr.aymanjawadi@gmail.com; 5Division of Sport Motoric and Methodology in Kinantropology, Department of Kinesiology, Faculty of Sport Science, Masaryk University in Brno, Kamenice 753/55, 62500 Brno, Czech Republic; zvonar@fsps.muni.cz

**Keywords:** DDH, prevalence, differential diagnosis

## Abstract

(1) Background: The objective of this study was to assess the prevalence of Developmental Dysplasia of the Hip (DDH) as a primary or secondary diagnosis during physiotherapy practice. No other studies have investigated the prevalence and associations of DDH within the practice of pediatric rehabilitation. (2) Methods: This retrospective review was performed on 12,225 physiotherapy referrals to the King Abdullah Specialized Children’s Hospital (KASCH), Riyadh, Kingdom of Saudi Arabia, from May 2016 to October 2021. Only DDH referrals for conservative treatment were included in the study. The plan for brace treatment was carried out by the pediatric orthopedics clinic in KASCH. The diagnostic methods were either a pelvic radiograph or ultrasound, depending on the participant’s age. DDH is considered one of the most common secondary complications for children with other medical diagnoses. (3) Results: The most common indication for referral was neurological diagnosis (44%), followed by orthopedic (28%), genetic (19%), cardiac (5%), ophthalmologic (3%), dermatologic (1%) and rheumatologic (0.5%) diagnoses. (4) Conclusion: The prevalence of DDH among all referrals in this study was 6%. In physiotherapy practice, neurologic, genetic, and orthopedic primary or secondary diagnoses were the most prevalent when DDH referrals were investigated. A relatively high prevalence of DDH in the pediatric rehabilitation clinic at KASCH in Riyadh was reported in this study.

## 1. Background

Developmental Dysplasia of the Hip appears as a dysplastic disorder. It describes anomalies of articular and periarticular anatomy, and their effects on biomechanics, explaining the hip instability, capsular laxity, and abnormal growth of the acetabulum [1].

Understanding the description of DDH and the following spectrum of hip abnormalities requires expert knowledge of hip joint growth and development [2]. The development of the acetabulum and head of the femur are intimately related. Dysplasia of the hip may occur in utero, perinatally, postnatally, during infant age, or later in childhood [2,3].

DDH includes unstable hips, subluxation, dislocation (luxation), and/or malformed acetabula [2,4]. A hip joint is unstable when the tight space between the femoral head and the acetabulum is lacking. The unsteady femoral head can subluxate or dislocate the bounds of the acetabulum [2,4]. Contact failure between the head of the femur and the acetabulum causes the dislocation of the hip [2,4].

Canavese (2020) [5] stated that there is a lack of quality evidence to prepare clinical practice guidelines. The purpose of this study was to highlight the importance of physiotherapy outcomes in a variety of correlations between DDH and other diagnoses. No other studies have investigated the prevalence and associations of DDH within the practice of pediatric rehabilitation. Worldwide, physiotherapists and occupational therapists significantly contribute to DDH treatment not only in post-surgery cases but also in conservative treatment. Many patients are diagnosed with DDH later in life, with no or a very late rehabilitation intervention.

### 1.1. Etiology

The etiology of DDH in children with normal development is unknown. The development of this condition appears to be altered by genetic and environmental influences [6,7]. The progressive changes of the acetabulum are connected to changes within the position and shape of the femoral head [4].

The prevalence of DDH recorded in the literature varies. This is because of several factors, such as the time of examination, and the variability of investigations in initial clinical examinations or follow-up ultrasonography. The incidence rate for DDH ranges from 1–20/1000 or 1–34/1000 (cases per birth), depending on the source of information and region [8,9]. According to the latest publication from Sadat-Ali in 2020, the average frequency of DDH in the Kingdom of Saudi Arabia (KSA) is 10.46 cases per 1000 births [10]. This extensive range of prevalence may exist due to genetic predisposition and/or cultural practice within certain races/ethnicities [11]. Recent research has suggested that only a strong family history, breech presentation, and congenital talipes calcaneo-valgus are confirmed risk factors [12]. Another publication from 2019 [13] reported torticollis and oligohydramnios as contributing factors. According to LeBel (2005) [14], abnormal acetabular development may also be due to poor hip position in infant carriers.

### 1.2. Prevalence

One of the most common secondary complications for pediatric patients with other medical conditions is DDH [15,16,17]. The second most common musculoskeletal abnormality in children with cerebral palsy (CP) is hip displacement, especially in children with spastic quadriplegia [18]. Hip problems are reported for 25–75% of children with CP [19]. Pain sensitivity was recognized in 40–84% of CP cases [20]. A teratologic dislocation of the hip (TDH) occurs in association with several malformation syndromes (Ehlers–Danlos syndrome, Down’s syndrome, Larsen’s syndrome, and arthrogryposis). Neuromuscular dysplasia of the hip (NDH) appears as a result of various muscle tone abnormalities in the neurologic disorders spina bifida and cerebral palsy [15,16,17]. DDH appears among 0.1–5.2% of newborns [21]. Bialik (1998) [22], Chan (1997) [23] and Graf (2014) [24] consider the increased prevalence of DDH as one of the best indicators in detecting DDH as early as possible [22,23,24]. Undiagnosed or late-diagnosed cases lead to devastation for the patients, families, and health system [22,23,24]. Limping, leg length discrepancy, pain, frequent surgeries, femoral head necrosis, osteoarthritis, disability, and total hip replacement at young ages are results of hip dysplasias that were undiagnosed early on, left untreated, or treated inappropriately [22,23,24]. Expenses for hospitalization, surgeries, rehabilitation, and indirect costs caused by a limited ability to work are incredibly high due to DDH complications. The early investigation and treatment of DDH are important to provide the best clinical results [18,25,26].

### 1.3. Consequences of Dislocation

According to Nandhagopal (2021) [3], moderate to severe osteoarthritis is the result of undiagnosed hip dislocation. Other complications include back pain, knee pain, limb length discrepancy, and gait abnormalities [2,10,12]. It affects the bones: shallow acetabulum suppressed by an overgrowth of fibrocartilage with the restrain position of the femoral head to the dorsal ilium. The flattening of the femoral head is caused by delayed bone ossification [27]. The shortening and anteversion position of the femoral neck cause a reduction in the patella, which appears in the medial position due to the internal rotation [27]. The soft tissue deteriorates, leading to joint capsule changes and changes to the suspensory ligament of the pelvis to support the body weight, leading to the expected hypertrophy and stiffness [27]. Contraction of the hip adductors and hamstring muscles can contribute to difficulties in reducing the joint during surgery, which is required for a missed diagnosed case of established hip dislocation [27].

### 1.4. Conservative Treatment

The latest systematic review by Pavone, et al. (2021) [28] highlighted bracing as the gold standard for DDH treatment for patients under six months of age. They divide splints into dynamic and static types. A Pavlik harness is the most common dynamic splint. Others are the Tibungen splint, the Frejka pillow, the Von Rosen splint, the Aberdeen splint, Coxaflex and the Teufel brace. Static braces include the rhino brace, the Milgram brace, the Denis brown bar, and the Ilfeld harness. According to Pavones study, orthopedic practitioners prefer dynamic splints [28]. In this study, the choice of DDH treatment was prescribed by orthopedic physicians.

## 2. Materials and Methods

A retrospective study was completed for all physiotherapy referrals to the pediatric rehabilitation clinic at the King Abdullah Specialized Children’s Hospital (KASCH) in Riyadh, Kingdom of Saudi Arabia (KSA). KASCH is the first medical referral institution specializing in children’s health care in KSA. The first physiotherapy referral for abduction braces (AB) was carried out in May 2016 by the orthopedic clinic, and the data were collected from then until the end of October 2021.

The inclusion criterion was a physiotherapy referral for brace treatment ordered by the pediatric orthopedic team in KASCH. All participants were accepted by the pediatric rehabilitation screener as category A, according to the screening procedure of the pediatric rehabilitation clinic in KASCH. They were seen as a walk-in the same day after an orthopedic appointment, by the principal investigator or by co-investigator, for fitting and family education. Exclusion criteria were physiotherapy referrals that did not include brace treatment for DDH or referrals for brace treatment from other clinics without assessment and appointment by the orthopedic clinic. After receiving referrals from other clinics for a conservative or brace treatment of DDH, the referring physician was contacted and informed to refer participants to the orthopedic clinic in KASCH for further assessment.

The pediatric orthopedic clinic in KASCH used pelvic radiographs or ultrasound for diagnostics. In KSA, both diagnostic methods were performed by expert musculoskeletal radiography technicians who were supervised by musculoskeletal radiology consultants, and the results were assessed by radiographic physicians. For participants less than three months of age, an ultrasound was used and for more than three months of age, a pelvic radiograph was used. Following these results, the orthopedic physician chose the treatment options for DDH. To assess acetabular development, the acetabular index was measured. The tolerable acetabular index (AI) is 27.5° in newborns, and at two years of age it should be 20° or less [11]. Rehabilitation screening sheets were utilized as the main resources for filtering referrals. Primary and secondary DDH cases receiving treatment with abduction braces and Pavlik harnesses were included. This retrospective study was important for further project developments. Data were collected by the principal investigator and co-investigator. From electronic health record (EHR), demographic information (for example: gender, date of birth, date of referral, etc.) were collected by the principal investigator.

Upon completion, the data were entered, reviewed, and analyzed. Descriptive statistics will be presented as frequencies and percentages (%). Statistical analysis was carried out using Statistical Package for Social Science version 24 (IBM Corp., Armonk, NY, USA).

## 3. Results

From May 2016 to the 30th of October 2021, 12,225 physiotherapy referrals prescribed by the multidisciplinary team were received, from which 678 (6%) were for brace fitting (Table 1). Among those patients, 529 (78%) were female and 149 (22%) male, with an average age of 6.7 + 0.3 months. The youngest patient for brace treatment was a 13-day-old male referred in 2018. The oldest patient was a nine-year-old female with agenesis of corpus callosum referred in 2021. A total of 187 (28%) participants had a positive family history. In total, 257 participants (38%) were delivered normally while the others had Cesarean deliveries (62%). The number of participants with breech presentation totaled 152 (23%), while participants with vertex position totaled 523 (77%) (Table 2). The pediatric orthopedic clinic in KASCH used pelvic radiographs or ultrasound for diagnostics. The median for pelvic radiographs evaluation of the acetabular index in the right hip was 28° (*n* = 582), with a minimum of 16° and a maximum of 49°. For the left hip acetabular index median was 29° with a minimum of 14° and a maximum of 52° (*n* = 594). Ultrasound was completed for 33 left hips with a median of 57%, where minimum coverage was 24% and maximum coverage was 69%. At the right hip, the median was 60% (*n* = 33), minimum coverage was 40%, and maximum coverage was 71%. Only coverage percentage was included in the study but some orthopedic and radiographic result notes were missing alpha angles in degrees.

### 3.1. Prevalence with Other Diagnoses

Idiopathic DDH showed in 454 (67%) participants and 224 (33%) had DDH secondary to another diagnosis, before or during brace treatment, as shown in Table 3 and Figure 1. This was presented in a combination of diagnoses from genetic, neurologic with orthopedic, and cardiology, and vice versa. Hypotonia was presented in 39 patients (17%), spasticity in 9 (4%), and global developmental delay in 24 (11%).

#### Correlated Diagnoses

Below are the listed diagnoses that were present in more than five participants (2%).

Neurologic and genetic diagnoses were: hypoxic-ischemic encephalopathy (HIE), agenesis of the corpus callosum, cerebral parenchymal hemorrhage, subgaleal hematoma, bilateral germinal matrix hemorrhage, spina bifida, brachial plexus, microcephaly, hydrocephalus with a ventriculoperitoneal shunt, Down’s syndrome, muscular dystrophy and spinal muscular atrophy, cleft palate, CHARGE syndrome, Costello syndrome, arthrogryposis, Joubert syndrome, Koolen de Vries syndrome, Frank Ter Haar syndrome, Kaufman syndrome, Pompe disease, Warkany syndrome, Cockayne syndrome, moya-moya, Klinefelter syndrome, and increased or decreased gene Glucose-6-phosphate dehydrogenase (G6PD).

Musculoskeletal disorders were: scoliosis, metatarsal adductor, talipes calcaneovalgus, calcaneovarus and talipes equinovarus (TEV), Perth’s disease, status post femur or tibia fracture, osteopenia and Rickets syndrome, infant torticollis, clinodactyly, congenital short femur, and overlapping toes.

Other factors for DDH included: conjoined twins post-separation, preterm 27–36 weeks in 38 participants (17%), twins in 17 (8%), triplets, respiratory distress syndrome (RDS) post-delivery in 25 (11%), and oligohydramnios.

Cardiac, ophthalmology, dermatology, and rheumatology diagnoses were present in fewer than five participants (<2%): atrial septal defect (ASD), patent ductus arteriosus (PDA), congenital heart disease, ductus arteriosus, strabismus, pendular nystagmus, micro cornea syndrome, optic disc coloboma, severe eczema, and Bechet disease.

## 4. Discussion

This study presents information regarding the importance of the early diagnosis of DDH in physiotherapy practice. It also had the limitation that only referrals for brace or conservative treatment provided by orthopedic clinic are included in the study.

The epidemiology and etiology of DDH are unknown due to the lack of standardization criteria in various publications [29,30]. There are many studies carried out by physicians and those publications evaluate prevalence with a specific diagnosis from their point of view. Loder and Skopelja (2011) [29] conducted a systematic review, focused on etiology, epidemiology, and diagnosis, and their published results excluded rehabilitation, surgery, and therapy. Their conclusion was: “That epidemiology of DDH is changing according to archeological studies” [29]. Goiano, Akkari, Pupin, and Santili (2020) [30] conducted their study on males only. A retrospective cohort review conducted by Pollet, Percy, and Prior (2017) [9], rules out the prevalence and the approximate risk factors of DDH in the population without screening. There is no DDH screening program in KSA as in Canada [9]. Well-described neonatal screening programs exist in Europe and Australia [9]. The assessments and diagnostics of possible hip deformities, such as abduction limitation and instability hip tests such as Ortolani and Barlow, depend on health providers at well-baby clinics [9,31]. The studies of Polet (2017) [9] and Wiliams (2018) [31] suggested that a standard clinical sonographic screening of high-risk communities decreased the rate of late presentation and the need of surgical intervention later in life [9,31]. This study shows that females with negative family history in vertex presentation, with the associated diagnosis, are at a higher risk of developing DDH.

Breech presentation during pregnancy increases the incidence of DDH. Loder, and Skopelja, in 2011 [29], published an article that reviewed data from multiple countries. Their conclusion was that breech presentation in a fetus with DDH ranges from 7.1% to 40%. In KASCH, Riyadh, KSA 38% of patients with DDH were breech and 8.8% were in the vertex position [29]. Within this study, 23% of patients were presented in the breech position. They described the type of breech presentation as very important. Their publication mentions frank-breech, hip flexion, knee extension, non-frank, or other one-side hip and knee flexion [29]. However, in this study, the results were not able to be compared due to EHR restrictions.

A positive family history led to 28% of participants from KASCH, whereas Loder and Skopelja only mentioned 21% in KSA [29]. They suggest an incidence of 3.4 pediatric patients <37 weeks maturation age [29], whereas this study identified the incidence for preterm 27–36 weeks was 5.6. As there are no previous studies conducted in KSA, the verification of the occurrence within the results is limited. Kural et al. (2019) [13] mention in their publication inconsistent results in the literature about oligohydramnios as risk factors of DDH. Similar to this study, oligohydramnios was not found as a risk factor as it was detected in one patient only.

According to Swarup, Penny, and Dodwell, the (2018) [11] coverage of the femoral head by the acetabulum should be at least 50% to be considered normal. In this study, it was 60% coverage for the right hip and 57% coverage for the left hip, which are considered normal. However, a limitation was the missing alpha angles degrees in some results [11]. They also mention radiographs, stating that “Atypical radiographic results may confirm the diagnosis of DDH, but a regular result of radiograph does not exclude instability” [11]. To determine acetabular development, orthopedists and radiologists use the acetabular index. By two years of age, the AI value should be less than 20° and after delivery, it is 27.5 [11]. Participants in this study had AI 28° in the right hip with a maximum of 49°; for the left hip it was 29° and the maximum AI was 49°. The results for this study indicate that the left hip is at more risk of developing DDH than the right hip, following a radiographic assessment [11].

The prevalence of DDH with an associated diagnosis in this study was 33%, including DDH referrals for brace treatment, which may be a limitation of this study.

This review is part of a dissertation study that focuses on the impact of DDH on gait, heel position, and feet pronation. As mentioned by Kasovic, 2020 [32]: “…no study has systematically settled factors associated with several foot functions in children…”. Gimunova, 2022 [33], discussed in their study the possible limitation that: “Future projects focusing on the pediatric developmental rate until the child reaches the higher gross motor milestone such as standing or walking alone. New kinematic and dynamic gait results will bring more understanding into gait evolution”.

After analyzing the prevalence and association, it is obvious that patients with multiple diagnoses had a high prevalence of developmental hip dysplasia. These were patients who were referred for braces therapy only. However, a review of all physiotherapy referrals to assess the association between other diagnoses and DDH in rehabilitation practice would be the most appropriate next step.

Coordination between the rehabilitation department and other clinics is very important. Early physiotherapy leads to a reduction in the rehabilitation duration, providing important results in terms of the acetabular angles, and an improvement in the general functionality as well. The dedication, consistency, and involvement of the multidisciplinary team should focus on the prevention and general improvement of the patient. The observations of parents are the most important element in the initial identification of the functional limitations in the first days after birth. Parents are also important for the therapeutic intervention, as valuable members of the team, along with the physician and the physical therapist.

We recommend that further studies include all participants with developmental dysplasia of the hip, primary, secondary or after surgery, including those referred by other clinics as well. One of the further projects is to create a DDH screening tool for all patients at rehabilitation practice in KASCH, to prevent further complications in patient improvement and during physiotherapy treatment.

## 5. Conclusions

The prevalence of DDH among all referrals in this study was 6%. In physiotherapy practice, neurologic, genetic, and orthopedic primary or secondary diagnoses were the most prevalent when DDH referrals were investigated. A relatively high prevalence of DDH in the pediatric rehabilitation clinic at KASCH in Riyadh was reported in this study.

## Figures and Tables

**Figure 1 children-09-00247-f001:**
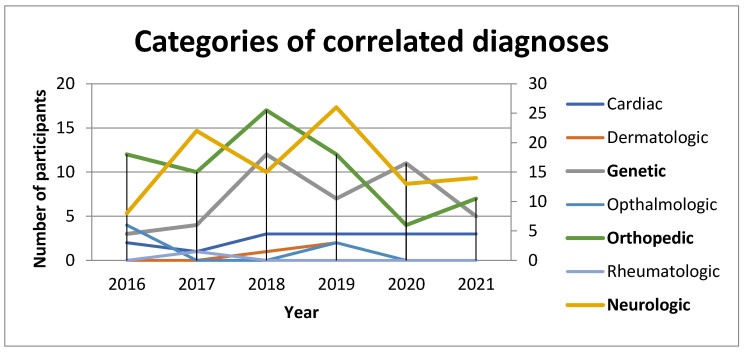
Categorization of diagnoses.

**Table 1 children-09-00247-t001:** Prevalence of DDH within physiotherapy referrals.

Year	Number of New Physiotherapy Referrals	Number of Physiotherapy Referrals for DDH Bracing (%)
2016	1058	95 (9%)
2017	1779	109 (6%)
2018	1679	157 (9%)
2019	2632	149 (6%)
2020	2512	82 (3%)
2021	2565	86 (3%)
Total	12,225	678 (6%)

**Table 2 children-09-00247-t002:** Demographic information.

Demographic Information	Categories	Total Sample (%, Frequencies)
Gender	Male	149 (22%)
	Female	529 (78%)
Family history	Negative	488 (72%)
	Positive	187 (28%)
Child position during pregnancy/delivery	Vertex	523 (77%)
	Breech	152 (23%)
Delivery procedure	Normal delivery	418 (62%)
	Cesarean delivery	257 (38%)
Participant age	Male	7.8 + 0.7
	Female	6.5 + 0.3

**Table 3 children-09-00247-t003:** Correlation of DDH with other diagnoses.

Diagnoses	Categories	Total Sample (%)	
Only DDH		454 (67%)	
Associated diagnoses with DDH	Neurologic	98 (44%)	224 (33%)
	Orthopedic	62 (28%)	
	Genetic	42 (19%)	
	Cardiac	12 (5%)	
	Ophthalmologic	6 (3%)	
	Dermatologic	3 (1%)	
	Rheumatologic	1 (0.5%)	

## Data Availability

Not applicable.

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
