# Peer review of "Developmental Dysplasia of the Hip: Prevalence and Correlation with Other Diagnoses in Physiotherapy Practice—A 5-Year Retrospective Review"

_children, 2022, doi:10.3390/children9020247_

Round 1

Reviewer 1 Report

The manuscript is retrospective study aimed to assess the prevalence of Developmental 22 Dysplasia of the Hip (DDH) as primary or secondary diagnosis during physiotherapy practice. A retrospective review was performed on 12,225 physiotherapy referrals in King Abdullah Specialized Children Hospital (KASCH) in Riyadh, Kingdom of Saudi Arabia, from May 2016 until October 2021. Separation of DDH referrals for brace treatment was done. The plan for brace treatment was carried out by the Pediatric orthopedics clinic in KASCH, for diagnostic pelvic radiograph or ultrasound was used according to participant age. DDH is considered as one of the most common secondary complications for children with other medical diagnosis.

I read the article with interest, the title is well thought out and faithfully reflects the content of the study.

  1. The abstract is sufficiently developed, and it is useful to frame the purpose of the study, but a few concerns are present:

Comment 1: It would be appropriate not to specify the limits of the study in the abstract.

  1. In the introduction, the characteristics of the DDH have been sufficiently described.

Comment 2: “The acronym DDH includes hips that are unstable, subluxated, dislocated (luxated), and/or have malformed acetabula. A hip is unstable when the tight fit between the femoral head and the acetabulum is lost, and the femoral head is able to move within (subluxated) or outside (dislocated) the confines of the acetabulum.” Please adding some bibliographic references about it, for example: (Canavese F. et al (2020) " Developmental dysplasia of the hip: Promoting global exchanges to enable understanding the disease and improve patient care").

Comment 3: For the sake of completeness, it would be advisable to add some information regarding the conservative management of DDH. Adding some bibliographic references about it, for example: (Pavone V. et al (2021) " Dynamic and Static Splinting for Treatment of Developmental Dysplasia of the Hip: A Systematic Review").

  1. The materials and methods have been shortly developed.

Comment 4: The inclusion and exclusion criteria for the study should be clarified.

Comment 5: Did they all have the same month of life when DDH was diagnosed?

Comment 6: Who did the ultrasound? Was he a pediatric orthopedist?

Comment 7: “KASCH’s pediatric orthopaedic clinic used pelvic or ultra106 radiographs for diagnostics.” How did you select who had to do the X-ray and who had to do the ultrasound?

  1. The discussion is sufficiently developed.

Comment 8: You did not specify the limitations of your study and what could be done in subsequent studies on the topic to overcome the limitations of this study.

Comment 9: The aim of your study should not be included again in the conclusions.

Finally, English language editing is needed.

Author Response

Response to Reviewer 1

I read the article with interest, the title is well thought out and faithfully reflects the content of the study.

 Dear reviewer 1. Thank you very much for the valuable comments. We update the article according to your suggestions. All changes were done with blue color for your follow-up.

The abstract is sufficiently developed, and it is useful to frame the purpose of the study, but a few concerns are present:

Comment 1: It would be appropriate not to specify the limits of the study in the abstract.

Dear reviewer 1. Thank you very much for the valuable comment. The sentence mentioning the limitations of the study was deleted.

In the introduction, the characteristics of the DDH have been sufficiently described.

 Comment 2: “The acronym DDH includes hips that are unstable, subluxated, dislocated (luxated), and/or have malformed acetabula. A hip is unstable when the tight fit between the femoral head and the acetabulum is lost, and the femoral head is able to move within (subluxated) or outside (dislocated) the confines of the acetabulum.” Please add some bibliographic references about it, for example: (Canavese F. et al (2020) " Developmental dysplasia of the hip: Promoting global exchanges to enable understanding the disease and improve patient care").

Dear reviewer 1. Thank you very much for the valuable comment. Bibliographic reference was added according to your suggestion.

…Canavese (2020) [5] stated that there is a lack of quality evidence to prepare clinical practice guidelines...

 Comment 3: For the sake of completeness, it would be advisable to add some information regarding the conservative management of DDH. Adding some bibliographic references about it, for example: (Pavone V. et al (2021) " Dynamic and Static Splinting for Treatment of Developmental Dysplasia of the Hip: A Systematic Review").

 Dear reviewer 1. Thank you very much for the valuable comment. Information regarding conservative treatment was implemented into the article:

The latest systematic review by Pavone, et al. (2021) [28] highlighted bracing as the gold standard for DDH treatment under six months of age. They divide splints into dynamic and static. Pavlik harness is the most common dynamic splint. Others are Tibungen splint, Frejka pillow, Von Rosen splint, Aberdeen splint, Coxaflex and Teaufel brace. Static braces are the rhino brace, Milgram brace, Denis brown bar, and the Ilfeld harness. According to Pavone's study orthopedics prefer dynamic splints.[28] In this study, the choice of DDH treatment was prescribed by orthopedic physicians.

The materials and methods have been shortly developed.

Comment 4: The inclusion and exclusion criteria for the study should be clarified.

 Dear reviewer 1. Thank you very much for the valuable comment. Both criteria were specified:

Including criteria were physiotherapy referrals for brace treatment ordered by the pediatric orthopedic team in KASCH. All participants were accepted by the pediatric rehabilitation screener as category A, according to the screening procedure of the pediatric rehabilitation clinic in KASCH. They were seen as a walk-in the same day after an orthopedic appointment, by the principal investigator or by co-investigator for fitting and family education. Exclusion criteria were all physiotherapy referrals that did not include brace treatment for DDH or referrals for brace treatment from other clinics without assessment and appointment by the orthopedic clinic. After receiving referrals from other clinics for conservative or brace treatment of DDH, the referring physician was contacted and informed to refer participants to the orthopedic clinic in KASCH for further assessment.

Comment 5: Did they all have the same month of life when DDH was diagnosed?

 Dear reviewer 1. Thank you very much for the valuable comment. The average age was 6.7 +0.3 as mentioned in the results. This is the average age when participants were referred to physiotherapy practice from an orthopedic clinic. The youngest and oldest ages of receiving patients were added:

The youngest patient for brace treatment was a 13 days old male referred in 2018. The oldest patient was a nine years old female with agenesis of corpus callosum referred in 2021.

Comment 6: Who did the ultrasound? Was he a pediatric orthopedist?

 Dear reviewer 1. Thank you very much for the valuable comment. The specifics regarding diagnostics were added into the article:

In KSA both diagnostic methods were performed by expert musculoskeletal radiography technicians that were supervised by musculoskeletal radiology consultants and the results were assessed by radiographic physicians. For participants less than three months of age an ultrasound was used and for more than three months of age, a pelvic radiograph was used.

Comment 7: “KASCH’s pediatric orthopedic clinic used pelvic or ultra106 radiographs for diagnostics.” How did you select who had to do the X-ray and who had to do the ultrasound?

 Dear reviewer 1. Thank you very much for the valuable comment. The specifics regarding diagnostics were added to the article. Can you please specify ultra106? We discussed ultra106 with an orthopedic consultant and he never heard about it.

The discussion is sufficiently developed.

Comment 8: You did not specify the limitations of your study and what could be done in subsequent studies on the topic to overcome the limitations of this study.

Dear reviewer 1. Thank you very much for the valuable comment. It was updated in the discussion:

After analyzing the prevalence and association, it is obvious that patients with multiple diagnoses had a high prevalence of developing developmental hip dysplasia. These were patients referred for braces therapy only. However, a review of all physiotherapy referrals to assess the association between other diagnoses and DDH in rehabilitation practice would be the most appropriate next step. 

Comment 9: The aim of your study should not be included again in the conclusions.

 Dear reviewer 1. Thank you very much for the valuable comment. The aim was deleted from conclusions:

This study is presenting information regarding the importance of early diagnostics of DDH in physiotherapy practice. With the limitation of including only referrals for a brace or conservative treatment provided by an orthopedic clinic. Further studies are recommended to include all participants with developmental dysplasia of the hip, primary, secondary, or after surgery, referred by other clinics as well. One of the further projects is to create a DDH screening tool for all patients at rehabilitation practice in KASCH to prevent further complications in the patient improvement and during physiotherapy treatment.

Finally, English language editing is needed.

Dear reviewer 1. Thank you very much for the valuable comment. English editing was done with a native speaker after all modifications were finished.

Reviewer 2 Report

Great work on your hard work and this paper.

Vasilcova et al present on the prevalence of DDH amongst their physiotherapy team referrals and incidence of comorbid/demographic factors. 

  1. Title and the use of the word “associated/association” - this word implies there was a statistical or clinical association of factors with a treatment or diagnosis. It appears that no statistical association analysis was conducted and therefore to remove ambiguity the word should be replaced with the likes of “correlated”. Replacement should be advised for the title and in line 189 where it was misused.
  2. Introduction and background information was very extensive but unnecessarily long. Reducing it would be more beneficial and limiting it to why this paper is helpful would be more focused.
  3. There is no highlighted purpose or objective in the introduction/background section. Please insert 
  4. Minor phrasing english editing required in the discussion example line 213, line 192 
  5. conclusion - the second paragraph should be apart of the discussion. This paper did not support or show results for such large conclusions and should be moved. 

Author Response

Response to Reviewer 2

Great work on your hard work and this paper.

Dear reviewer 1. Thank you very much for the valuable comments. We update the article according to your suggestions. All changes were done with green color for your follow-up.

Vasilcova et al present the prevalence of DDH amongst their physiotherapy team referrals and incidence of comorbid/demographic factors. 

Title and the use of the word “associated/association” - this word implies there was a statistical or clinical association of factors with treatment or diagnosis. It appears that no statistical association analysis was conducted and therefore to remove ambiguity the word should be replaced with the likes of “correlated”. Replacement should be advised for the title and in line 189 where it was misused.

Dear reviewer 2. Thank you very much for the valuable comment. The title was adjusted accordingly:

Developmental Dysplasia of the Hip: Prevalence and Correlation with other Diagnoses at Physiotherapy Practice: 5 years retrospective review.

The introduction and background information was very extensive but unnecessarily long. Reducing it would be more beneficial and limiting it to why this paper is helpful would be more focused.

Dear reviewer 2. Thank you very much for the valuable comment. It was taken into consideration. Information in the background is necessary. Etiology and Prevalence are highlighting the correlation of DDH with other diagnoses and the consequence. As per professional experience, especially in KSA, that information is necessary for readers (therapists, physicians, others) to understand the importance of screening for DDH. 

There is no highlighted purpose or objective in the introduction/background section. Please insert

Dear reviewer 2. Thank you very much for the valuable comment. The purpose of this study was inserted.

The purpose of this study was to highlight the importance of physiotherapy outcomes in a variety of correlations between DDH and other diagnoses. There are no other studies investigating the prevalence and associations of DDH within the pediatric rehabilitation practice. Worldwide, physiotherapists and occupational therapists significantly contribute to DDH treatment not only in post-surgery cases but in conservative treatment as well. Many patients are diagnosed with DDH later in age with non or very late rehabilitation intervention. 

Minor phrasing English editing required in the discussion example line 213, line 192 

Dear reviewer 2. Thank you very much for the valuable comment. The rephrasing and English editing were done with a native speaker.

conclusion - the second paragraph should be a part of the discussion. This paper did not support or show results for such large conclusions and should be moved. 

Dear reviewer 2. Thank you very much for the valuable comment. The paragraph was moved into the discussion:

Coordination between the rehabilitation department and other clinics is very important. Early physiotherapy is leading to reduction of the rehabilitation duration, reaching important results of the acetabular angles, and an improvement of the general functionality as well. The dedication, consistency, and involvement of the multidisciplinary team should focus on prevention and general improvement of the patient. Parents are the most important part in the identification of the functional limitations from the first days of birth. Parents are also important for the therapeutic intervention, within the time they are becoming a member of the team, along with the physician and the physical therapist.